# Inhibition of the Akt/PKB Kinase Increases Na_v_1.6-Mediated Currents and Neuronal Excitability in CA1 Hippocampal Pyramidal Neurons

**DOI:** 10.3390/ijms23031700

**Published:** 2022-02-01

**Authors:** Mate Marosi, Miroslav N. Nenov, Jessica Di Re, Nolan M. Dvorak, Musaad Alshammari, Fernanda Laezza

**Affiliations:** 1Department of Pharmacology & Toxicology, The University of Texas Medical Branch, Galveston, TX 77555, USA; mgmarosi@utmb.edu (M.M.); nenovmir@gmail.com (M.N.N.); jedire@utmb.edu (J.D.R.); nmdvorak@utmb.edu (N.M.D.); malshammari@ksu.edu.sa (M.A.); 2Department of Pharmacology, College of Pharmacy, King Saud University, Riyadh P.O. Box 145111, Saudi Arabia; 3Center for Addiction Research, Center for Biomedical Engineering and Mitchell, Center for Neurodegenerative Diseases, The University of Texas Medical Branch, 301 University Boulevard, Galveston, TX 77555, USA

**Keywords:** protein kinase B (PKB)/Akt, voltage-gated sodium channels, electrophysiology, neuronal excitability, axon initial segment, immunohistochemistry

## Abstract

In neurons, changes in Akt activity have been detected in response to the stimulation of transmembrane receptors. However, the mechanisms that lead to changes in neuronal function upon Akt inhibition are still poorly understood. In the present study, we interrogate how Akt inhibition could affect the activity of the neuronal Na_v_ channels with while impacting intrinsic excitability. To that end, we employed voltage-clamp electrophysiological recordings in heterologous cells expressing the Na_v_1.6 channel isoform and in hippocampal CA1 pyramidal neurons in the presence of triciribine, an inhibitor of Akt. We showed that in both systems, Akt inhibition resulted in a potentiation of peak transient Na+ current (*I*_Na_) density. Akt inhibition correspondingly led to an increase in the action potential firing of the CA1 pyramidal neurons that was accompanied by a decrease in the action potential current threshold. Complementary confocal analysis in the CA1 pyramidal neurons showed that the inhibition of Akt is associated with the lengthening of Na_v_1.6 fluorescent intensity along the axonal initial segment (AIS), providing a mechanism for augmented neuronal excitability. Taken together, these findings provide evidence that Akt-mediated signal transduction might affect neuronal excitability in a Na_v_1.6-dependent manner.

## 1. Introduction

The serine/threonine kinase Akt (protein kinase B (PKB)) is a ubiquitously expressed enzyme that plays essential roles in cell growth, survival, motility, and regulation of metabolism, and homeostasis in almost all organs including, the brain [1,2,3,4,5,6,7,8,9]. In mammals, there are three Akt isoforms: Akt1, Akt2, and Akt3 (PKBα, PKBβ, and PKBγ, respectively) [10,11], which are highly expressed in the nervous system. Pertinent to its function as a prominent signaling molecule, inactive Akt is located in the cytosol and is recruited to the plasma membrane upon the activation of transmembrane receptors, such as tyrosine kinases receptors (TYK-R) and G-protein coupled receptors (GPCR) through phosphatidylinositol-3,4,5-trisphosphate (PIP_3_) [12,13,14,15,16,17,18]. Once Akt translocates to the cell membrane, it is activated by a multistep phosphorylation process that results in phosphorylation of Thr308 by phosphoinositide-dependent kinase 1 (PDK1) and Ser473 by phosphoinositide-dependent kinase 2 (PDK2) [17,19,20,21,22]. After activation, Akt phosphorylates several targets [7,23] at the consensus motif RXRXXS/T [24], including several intracellular signaling molecules, such as GSK3 [25,26,27] and tuberous sclerosis complex 2 [28]. Given the diverse downstream regulatory effects of Akt upon TYK-R and GPCR activation, it is unsurprising that Akt is involved in virtually all cellular functions.

In addition to the aforementioned regulatory effects of Akt throughout the body, the kinase has further been shown to confer important regulatory effects on several intrinsic neuronal properties in developing and adult neurons. For example, the activation of the AKT/mTOR pathway (active, phosphorylated Akt) is essential for neuronal development, neuronal cell survival [29], neurite outgrowth and branching, axonal formation, cytoskeleton organization [30,31,32,33,34,35,36], formation of synapses [32,37], neuronal plasticity and memory [38,39,40,41], and neurodegeneration [42,43,44,45]. Despite these known regulatory effects of Akt on central nervous system CNS neurons, the ultimate molecular targets that are acted upon by Akt to exert these modulatory effects remain poorly understood.

Given their canonical role in CNS signaling, ion channels stand out as plausible candidates that could be acted upon by Akt to exert regulatory effects on intrinsic neuronal properties; however as potential downstream targets of Akt, ion channels have only been marginally investigated in the brain. On account of their central role in initiating the action potential, the regulation of voltage-gated Na+ (Na_v_) channels by Akt would be of particular consequence, as small changes in the activity of these ion channels can corrupt circuits and can lead to the unbalanced function of neuronal activity and aberrant behavior outputs. Posttranslational modifications such as phosphorylation play an essential role in regulating the function of Na_v_ channels and have profound effects on the kinetics, subcellular localization, and trafficking of the channel, altering intrinsic the excitability and activity-dependent plasticity of the neurons [46]. Apart from a demonstration of a link between Akt and Na_v_1.7 and Na_v_1.8 in dorsal root ganglion (DRG) neurons [47] and demonstration of Akt importance in the regulation of Na_v_1.1, which are critical for interneurons function [48], information on the relationship between Akt and Na_v_ channels in the primary CNS neurons that are known to abundantly express the Na_v_1.2 and Na_v_1.6 channels is scarce. Providing evidence for a linkage between Akt and the Na_v_ channel, we have previously elucidated a complex kinase network that involves a diverse array of proteins related to Akt that regulates the macromolecular complex of the Na_v_ channel [49,50,51,52,53]. Pertinent to the present investigation, we have previously shown that the genetic and pharmacological manipulation of glycogen synthase kinase 3 (GSK3), a kinase whose activity is decreased upon phosphorylation by Akt, confers changes in the activity of the Na_v_ channels and neuronal excitability [52]. Related to this, we have shown that the pharmacological manipulation of Wee1, a kinase whose activity potentially increases Akt activity, also confers changes in Na_v_ channel activity [49,54]. Despite these reported regulatory effects on Na_v_ channel activity conferred by manipulating the kinases that control Akt activity, the effects of directly altering Akt activity on Na_v_ channel activity have been less well described.

To elucidate the latter and while focusing on the Akt-mediated regulation of the Na_v_1.6 channel on account of it serving as one of the primary channel isoforms for CNS principal neurons, we employed a pharmacological inhibitor of Akt called triciribine in the present investigation to study how Akt inhibition affected Na_v_1.6 channel activity and neuronal excitability. In voltage-clamp studies, the application of triciribine was shown to potentiate the peak transient Na current (*I*_Na_) in heterologous cells expressing the Na_v_1.6 channel and in CA1 pyramidal neurons in hippocampal slices. Consistent with these voltage-clamp recordings, we correspondingly demonstrate that the pharmacological inhibition of Akt increases the excitability of CA1 pyramidal neurons. Using confocal microscopy, we additionally show that the observed changes in CA1 pyramidal neuron activity are accompanied by changes in Na_v_1.6 pattern distribution at the axonal initial segment (AIS). When considered collectively, these results, in tandem with the results of previous investigations, demonstrate that the direct and indirect manipulation of Akt activity confers changes in Na_v_ channel and neural activity, which has important implications for unraveling the complex signaling cascades that fine-tune neuronal excitability.

## 2. Results

### 2.1. Effects of Pharmacological Inhibition of Akt on Na_v_1.6-Encoded Currents in In Vitro HEK-Na_v_1.6 Cells

To evaluate the effect of the Akt pathway on Na_v_1.6-encoded currents, HEK293 cells that were stably expressing the human Na_v_1.6 channel (HEK-Na_v_1.6) were treated with either vehicle (0.1% DMSO) or triciribine (30 µM) for 60 min prior to performing whole-cell patch clamp recording. After forming a stable whole cell patch-clamp configuration in a submerged recording chamber, cells were subjected to a series of depolarizing pulses to evoke inward Na^+^ currents (Figure 1A).

In the cells that had been pretreated with triciribine, we observed an upregulation of Na_v_1.6-encoded currents that was evident throughout the majority of the current–voltage relationship (from −20 to +40 mV) (Figure 1B). On average, we found that the Na_v_1.6-mediated peak current density measured at -10 mV (Figure 1C) was significantly higher (−99.4 ± 14.6 pA/pF, *n* = 11, *p* = 0.0076) in the triciribine (30 µM) -treated HEK-Na_v_1.6 cells compared to in the DMSO control (−47.8 ± 9.4 pA/pF, *n* = 11) (Figure 1C). The voltage-dependence of the activation and of steady-state inactivation of Na_v_ channels are critical parameters in determining intrinsic firing that affects the channel availability and the threshold of action potential. Thus, further analyses were performed to determine any potential effect of triciribine on Na_v_1.6 channel kinetics (Figure 1D,E). As shown in Figure 1D, the Na_v_1.6 currents in the presence of triciribine exhibited an activation profile that was indistinguishable from the control group (*p* = 0.96). The V_1/2_ of activation upon treatment with triciribine was −23.5 ± 2.1 mV (*n* = 10) compared to −23.6 ± 1.2 mV in the DMSO control group (*n* = 9) (Table 1).

To determine whether triciribine modified the steady-state inactivation properties of the Na_v_1.6 channels, cells from the two experimental groups were subjected to a standard two-step protocol, including a variable pre-step conditioning pulse followed by a test pulse. We found that the cells that had been treated with triciribine exhibited a V_1/2_ of steady-state inactivation of −59.5 ± 3.1 mV (*n* = 8), which was not significantly different (*p* = 0.358) from the DMSO value of −62.9 ± 1.9 mV (*n* = 11) (Figure 1E and Table 1). Taken together, these data, which are summarized in Table 1., demonstrate that the inhibition of the Akt pathway leads to an effect on the peak current densities without modifying the basic kinetic properties of Na_v_1.6-mediated currents.

### 2.2. Effects of Pharmacological Inhibition of Akt on Na_v_1.6-Encoded Currents and Neuronal Excitability in Hippocampal Pyramidal Neurons

To recapitulate the findings observed in the heterologous cells in neurons, whole-cell voltage-clamp recordings of *I*_Na_ were performed in intact hippocampal CA1 pyramidal neurons in the acute brain slice preparation using the voltage-clamp protocol shown in Figure 2A.

As described by Milescu et al. [55], a depolarizing pre-pulse step was used to inactivate Na_v_ channels distant from the recording electrode, and a second step was applied shortly afterward to record the Na_v_ channels close to the recording pipette (Figure 2A), similar to our previous publication [56]. Importantly, hippocampal CA1 pyramidal neurons represent a promising cellular target to be affected by changes in Akt signaling, as these cells abundantly express Na_v_1.6 channels where they contribute to somatic *I*_Na_ and intrinsic excitability, Na_v_1.6 channels are abundantly expressed in hippocampal CA1 pyramidal neurons, where they contribute to somatic *I*_Na_ and intrinsic excitability [57,58,59,60].

CA1 pyramidal neurons from slices treated with vehicle (DMSO) displayed an average peak *I*_Na_ density for the second test pulse of −49.5 ± 4.5 pA/pF (*n* = 8), whereas 20 µM triciribine treatment (60 min) induced a significantly increased peak *I*_Na_ current density of −99.4 ± 9 pA/pF (*n* = 10; *p* < 0.001; Figure 2B,C). Finally, we also measured the voltage-dependence of the activation and of steady-state inactivation of Na_v_ channels in CA1 pyramidal neurons. Our findings are consistent with the effect of triciribine treatment that we observed in heterologous cells expressing Na_v_1.6, namely that we did not find significant differences in these parameters between the Akt inhibitor (triciribine) treated and control groups (Figure 2D,E and Table 2). Taken together, these results suggest that Akt inhibition might increase neuronal excitability in CA1 neurons through Na_v_1.6 channel modulation.

To test whether the pharmacological inhibition of Akt affected the excitability of the CA1 pyramidal neurons, acute hippocampal slices were prepared and treated with either triciribine (20 µM) or its vehicle (DMSO) for 60 min, after which whole-cell patch-clamp recordings were performed. Whole-cell current-clamp recordings were obtained from visually identified CA1 pyramidal neurons and voltage responses, and action potentials were evoked by 800 ms long (−20 to +210 pA, Δ = +10 pA) current injections. In these experiments, we found that the triciribine-treated group showed increased action potential discharge (Figure 3A–C) compared to the DMSO control group.

We then assessed whether the action potential current and voltage thresholds were altered by triciribine treatment, which were evaluated using the first measurable action potential induced by the above-mentioned current injection protocol. We found that the current threshold (the mean injected current required to evoke an action potential) was significantly lower in the triciribine-treated group 71.8 ± 12.5 pA (*n* = 11) compared to the DMSO control group (124 ± 17.1 pA (*n* = 10); *p* = 0.0221 with Student’s *t*-test) (Figure 3C and Table 3).

The voltage threshold appeared to be lower in the triciribine-treated group (−45.8 ± 2.4 mV, *n* = 11) compared to control cells (−41.8 ± 2.1, *n* = 10) but this difference was not statistically significant (*p* = 0.1517) (Figure 3D and Table 3). Triciribine had no effects on other passive intrinsic properties and input resistance of the recorded cells (Table 3). Collectively considered, the results of these voltage and current-clamp recordings, in combination with the results shown in Figure 1, suggest that the pharmacological inhibition of Akt increases the intrinsic excitability of CA1 pyramidal neurons by potentiating the activity of their constituent Na_v_ channels.

### 2.3. Pharmacological Inhibition of Akt with Triciribine Alters the Length of the Na_v_1.6 Immunofluorescence Labelling at the AIS

Changes in neuronal excitability can occur in response to the increased localization of Na_v_ channels as well as the relative enrichment of particular Na_v_ channel isoforms at the action potential initiation site, the AIS. Recently, it has also been shown that treatment with triciribine alters the fluorescent intensity of βIV spectrin, a critical constituent of the AIS [61], suggesting that the inhibition of Akt may alter pattern distribution and the expression of other components of the AIS. Thus, we hypothesized that treatment with triciribine could be accompanied by a change in the pattern expression and distribution of Na_v_1.6 channels, which were then enriched at the AIS in the hippocampal CA1 pyramidal neurons. To test this hypothesis, ex vivo brain slices containing the hippocampus were exposed to triciribine (20 µM) or vehicle (0.02% DMSO) for 2 h and were processed to determine the immunofluorescence analysis of the CA1 region (Figure 4).

Cell bodies were labeled with the nuclear marker DAPI (Figure 4A,D), with a guinea pig antibody against the neuronal marker NeuN (Figure 4B,E), with a mouse monoclonal antibody against AnkyrinG, used as an AIS marker (Figure 4C,F,H,K), and a mouse monoclonal antibody against Na_v_1.6 (Figure 4C,F,G,J). As previously shown in other studies, AnkyrinG is enriched at the AIS of neurons [62,63], while Na_v_1.6 is found in both the soma (Figure 4C,F) and the AIS (Figure 4G,I,J,L). Following treatment with triciribine or DMSO, the immunofluorescence of Na_v_1.6 in AnkG positive AIS was traced, revealing a significant increase in the length of the Na_v_1.6 staining after treatment with triciribine compared to DMSO control (*p* = 0.0104 as analyzed by a two-tailed nested *t*-test; Figure 4M). Conversely, the treatment with triciribine did not alter the fluorescent intensity of Na_v_1.6 at the AIS as measured by the analysis of the cross-section of the AIS (Figure 4I,L,N). Overall, these results demonstrate that triciribine causes an increase in the length of Na_v_1.6 staining in the AIS, which is in line with previous results showing that the elongation of the AIS is associated with an increase in neuronal excitability [64,65].

## 3. Discussion

Previous studies have characterized a complex kinase network that regulates the activity of the Na_v_1.6 channel macromolecular and resultantly modulates neuronal activity [50,52,61]. In this network, the phosphorylation of either the Na_v_1.6 channel or its auxiliary proteins by various kinases has been shown to exert powerful regulatory effects on macromolecular complex assembly of the Na_v_1.6 channel and its activity. Pertinent to the present investigation, we have shown that GSK3, a kinase whose activity is decreased through phosphorylation by Akt, directly phosphorylates the Thr1936 of the Na_v_1.6 channel [52]. Through this phosphorylation of the Na_v_1.6 channel, GSK3 is able to regulate the channel’s activity, which resultantly confers changes in the excitability of neurons in clinically relevant brain regions, such as the nucleus accumbens, part of the striatum. Given this linkage between the Na_v_1.6 channel and GSK3, coupled with the latter’s activity being regulated by Akt, we postulated a signaling axis involving Na_v_1.6, GSK3, and Akt. To further interrogate this postulated signaling cascade, we sought to characterize the effects of the direct pharmacological inhibition of Akt on the Na_v_1.6 channel activity and on neuronal excitability in the present study.

Therefore, we provided functional evidence that Akt activity modulates the peak *I*_Na_ density elicited by heterologous cells expressing the Na_v_1.6 channel. Specifically, the treatment of HEK-Na_v_1.6 cells with the Akt inhibitor triciribine increased their peak *I*_Na_ density relative to treatment with vehicle (DMSO) (Figure 1). Our results also showed evidence for the modulatory effect of Akt activity on the intrinsic firing properties of CA1 pyramidal neurons. The observed effect of triciribine was twofold: (1) it increased the maximum number of action potential firing; and (2) it decreased the current threshold (Figure 3). These effects of triciribine on the firing properties of the CA1 pyramidal neurons in tandem with the lack of effects on the passive electrical properties (Table 3) and the results observed in heterologous cells collectively point toward the pharmacological inhibition of Akt increasing neuronal excitability by increasing the number of available Na_v_ channels [52,66,67]. Lending further evidence to the Na_v_ channel modulation underlying triciribine’s potentiation of the excitability of CA1 pyramidal neurons, our voltage-clamp experiments performed in CA1 pyramidal neurons in the intact slice preparation (Figure 2 and Table 2) show that triciribine increases the transient *I*_Na_ of these cells, similar to the results observed in heterologous cells.

Previously, it has been shown that the activity of different kinases in the AIS or inhibition of these kinases have significant effects on the subcellular distribution of AIS proteins, which determines the shape and ion channel composition of AIS [59]. For instance, βIV spectrin is sensitive to kinase perturbation (Akt inhibition) at the AIS and the dendrites of primary hippocampal neurons [61]. Based on these findings, we complemented our electrophysiology studies with confocal imaging (immunohistochemistry) experiments to assess any potential changes in Na_v_1.6 pattern distribution related to the functional phenotypes we observed in the hippocampal slices. We found that the inhibition of Akt in pyramidal neurons of the CA1 region leads to a lengthening of the Na_v_1.6 fluorescent labelling along the AIS, which supports the reported augmented excitability phenotype observed in our electrophysiologic results from this cell type.

The combined findings of these electrophysiological and imaging studies further demonstrate that intracellular phosphorylation pathways have a direct effect on voltage-gated Na^+^ currents and that the manipulation of these kinase networks causes modifications in the neuronal activity. Recently, it was found that the activation of Akt can result in a robust decrease in the Na_v_1.1-mediated sodium currents accompanied by significant changes in the inactivation of the channel. The effect of Akt was attributed to its ability to directly phosphorylate the Na_v_1.1 channel [48]. Thus, the mechanism by which the pharmacological inhibition of Akt alters neuronal activity in our study could arise by directly preventing the kinase mediated phosphorylation of the Na_v_1.6 channel, an indirect effect on the phosphorylation of the Na_v_1.6 channel conferred by increasing the activity of GSK3, or a combination of these direct and indirect effects; however, future in vitro phosphorylation studies will be necessary to unequivocally address how the pharmacological inhibition of Akt alters the phosphorylation of the Na_v_1.6 channel.

Overall, by employing a combination of electrophysiology and immunohistochemistry, we have further elucidated the cellular signaling cascades that regulate Na_v_ channel activity and neuronal excitability. In particular, our results point toward Akt functioning as a central node in an AKT-GSK3-Na_v_1.6 network, and that this pathway plays an important role in fine-tuning neuronal excitability. Considering the important role of Akt-mediated signaling in a diverse array of CNS conditions, including schizophrenia [68,69,70], depression [71], Alzheimer’s disease [72,73], and Parkinson’s disease [74,75,76,77], our results have important implications for refining our understanding of the molecular determinants of complex neuropsychiatric and neurodegenerative disorders.

## 4. Materials and Methods

### 4.1. Chemicals

Triciribine (EMD Chemicals, San Diego, CA, USA) was dissolved in 100% DMSO (Sigma-Aldrich, St. Louis, MO, USA) to a working stock concentration of 20 mM, aliquoted, and stored at −20 °C. Triciribine was further diluted as needed for experimental purposes.

### 4.2. Animals

C57/BL6J mice (Jackson Laboratory (Bar Harbor, ME, USA)) were housed in the University of Texas Medical Branch, Galveston, TX, USA vivarium, which operates in compliance with the United States Department of Agriculture Animal Welfare Act, the NIH Guide for the Care and Use of Laboratory Animals, the American Association for Laboratory Animal Science (Memphis, TN, USA), and Institutional Animal Care and Use Committee approved the protocols (University of Texas Medical Branch, Galveston, TX, USA).

### 4.3. Cell Culture

All reagents were purchased from Sigma-Aldrich, St. Louis, MO, USA, unless noted otherwise. HEK-293 cells stably expressing human Na_v_1.6 (HEK-Na_v_1.6 cells) were maintained in a medium composed of equal volumes of DMEM and F12 (Invitrogen, Carlsbad, CA, USA) supplemented with 0.05% glucose, 0.5 mM pyruvate, 10% fetal bovine serum, 100 U/mL penicillin, 100 µg/mL streptomycin, and 500 μg/mL G418 (Invitrogen, Carlsbad, CA, USA) to ensure stable Na_v_1.6 expression. Cells were maintained at 37 °C with 5% CO_2_.

### 4.4. Electrophysiology

#### 4.4.1. Cell Cultures Electrophysiology—Voltage-Clamp

HEK-Na_v_1.6 cells were treated with either DMSO (0.1%) or triciribine (30 μM). Single-cell manual patch-clamp recordings in voltage-clamp configuration were performed at room temperature (20–22 °C) using an Axopatch 200B amplifier (Molecular Devices, Sunnyvale, CA, USA). Borosilicate glass pipettes with a resistance of 3–8 MΩ were made using a Narishige PP-83 vertical Micropipette Puller (Narishige International Inc., East Meadow, NY, USA).

Whole-cell voltage-clamp recordings in heterologous cell systems were performed as previously described [78,79]. HEK-Na_v_1.6 cells were dissociated and re-plated at low-densities onto glass coverslips. After allowing 2–3 to ensure adherence, coverslips were transferred to a recording chamber filled with the following extracellular solution: (in mM): 140 NaCl, 3 KCl, 1 MgCl_2_, 1 CaCl_2_, 10 HEPES, 10 glucose, pH: 7.3. For the control recordings, 0.15% DMSO was added to the extracellular solution. For recordings to assess the effects of pharmacological inhibition of Akt, 30 µM of triciribine was added to the extracellular solution. The concentration of DMSO was maintained at 0.15% in both groups. Cover slips were incubated in the extracellular solution containing either vehicle (DMSO) or 30 µM triciribine at room temperature for 60 min prior to the start of recordings. The pipette (intracellular) solution contained (in mM): 130 CH_3_O_3_SCs, 1 EGTA, 10 NaCl, 10 HEPES, pH: 7.3.

Data were only collected from cells forming a 1 GΩ or tighter seal at a holding potential of −70 mV. Membrane capacitance and series resistance were estimated by the dial settings on the amplifier. After break-in, capacitive transients and series resistances were carefully compensated by 70–80% at the beginning of every protocol. Cells exhibiting a series resistance of 20 MΩ or higher were excluded from the analysis. The data were acquired with pClamp/Clampex 7/9 (Molecular Devices, Sunnyvale, CA, USA) and digitized with a Digidata 1322A A/D converter (Molecular Devices, Sunnyvale, CA, USA) at a sampling rate of 20 kHz and filtered at 5 kHz prior to digitization and storage. All of the experimental parameters were controlled by Clampex 7/9 software (Molecular Devices).

Cells were maintained at −70 mV holding potential, and voltage-dependent inward currents were evoked by depolarizations to test potentials between −100 mV and +60 mV (Δ +5 mV) from a holding potential of −70 mV followed by a voltage pre-step pulse of −120 mV (Na_v_1.6) (Figure 1A—inset). Steady-state inactivation of the Na_v_ channels was measured with a two-pulse protocol. From the holding potential (−70 mV), cells were stepped to varying test potentials between −120 mV and 20 mV (pre-pulse) prior to a test pulse to −10 mV.

#### 4.4.2. Brain Slice Electrophysiology

Acute hippocampal horizontal slices (300 μm) from adult C57/BL6J wild type mice (Jackson Laboratory (Bar Harbor, ME, USA) were prepared as previously described [56,66,80]. Briefly, the mice were anesthetized with isoflurane (Baxter, Deerfield, IL, USA) and were quickly decapitated. The brains were rapidly removed and placed in cold (0–4 °C) oxygenated cutting solution containing (in mM) 72 Tris-HCl, 18 Tris-Base, 1.2 NaH_2_PO_4_, 2.5 KCl, 20 HEPES, 20 sucrose, 25 NaHCO_3_, 25 glucose, 10 MgSO_4_, 3 Na-pyruvate, 5 Na-ascorbate, and 0.5 CaCl_2_ (Sigma-Aldrich, St. Louis, MO, USA); 300–310 mOsm, pH 7.4. Hippocampal slices containing the CA1 region were cut with a vibratome (VT1200 S, Leica Microsystems, Wetzlar, Germany) in a continuously oxygenated (mixture of 95%/5% O_2_/CO_2_) ice-cold tris-based cutting solution. After cutting, slices were incubated in the same cutting solution at 32 °C for 15 min. Then, slices were transferred and kept in standard (recording) artificial cerebrospinal fluid (aCSF) (in mM: 124 NaCl, 3.2 KCl, 1 NaH_2_PO_4_, 26 NaHCO_3_, 1 MgCl_2_, 2 CaCl_2_, 10 glucose; pH = 7.4 and osmolarity = 300–310 mOsm) at 32 °C at least for 60 min. After this recovery period, the slices were incubated for 60 min in a chamber containing continuously oxygenated and 31 ℃ standard aCSF treated with either 0.01% DMSO or 20 µM triciribine before starting the recording.

For the whole cell patch-clamp recordings (performed in current-clamp mode), the prepared brain slices were transferred and submerged in a recording chamber and perfused with standard recording aCSF (see above), which was continuously oxygenated with 95% O_2_ and 5% CO_2_ (pH 7.4). The flow rate in the recording chamber was kept at 1.5 mL/min, and the bath temperature was maintained at 30–32 °C by an inline solution heater and temperature controller (TC-344B, Warner Instruments, Hamden, CT, USA). Additionally, 20 µM bicuculline; 20 µM NBQX; and 100 µM AP5 (Tocris, Bristol, UK) were applied to the recording solution to block synaptic activity. The whole-cell current-clamp recordings from visually identified pyramidal neurons were performed by using an Axopatch 200B or Multiclamp 700B amplifier (Molecular Devices, Sunnyvale, CA, USA). Borosilicate glass pipettes with a resistance of 3–6 MΩ were made using a Narishige PP-83 vertical Micropipette Puller (Narishige International Inc., Amityville, NY, USA). Current-clamp recording were performed with pipettes filled with internal solution containing (in mM): 145 K-gluconate, 2 MgCl_2_, 0.1 EGTA, 2 Na_2_ATP, and 10 HEPES (pH 7.2 with KOH; and osmolarity = 290 mOsm). The pipette potential was adjusted to zero and the pipette capacitance was compensated before seal formation. Membrane potentials were not corrected for the liquid junction potential.

Data were only collected from cells forming a 1 GΩ or tighter seal. The series resistance was carefully compensated at the beginning of every protocol (in voltage-clamp mode). The maximal series resistance we accepted was 25 MΩ. Data were acquired at 50 kHz and filtered at 2 kHz before digitization and storage. All of the experimental parameters were controlled using Clampex 9.2 software (Molecular Devices, Union City, CA, USA) interfaced to the electrophysiological equipment using Digidata 1320A or 1322A analog-to-digital interfaces (Molecular Devices). Voltage responses and action potentials were evoked by hyper and depolarizing current pulses (with a range of current injections from −20 pA to 210 pA with 800 ms pulses and a change in injected current of 10 pA between sweeps) from a starting holding membrane potential of −70 mV. Input–output relationships were plotted as the number of spikes against the given current step (Figure 3). Only spikes with overshoots were taken for analysis (as described in Scala, Nenov et al., 2018. [52]). In the whole-cell current-clamp recordings, *n* = 7 and *n* = 6 mice were allocated into the DMSO and triciribine-treated groups, respectively (*n* = number of cells as listed in the corresponding Figure 3). Finally, to confirm that the results seen in current-clamp recordings were mediated by changes in Nav channel activity, whole-cell voltage-clamp recordings of INa were performed (*n* = 3 mice were used (half of the slices were treated with triciribine parallel with DMSO as control treatment). On account of space clamp issues, recording fast-gating *I*_Na_ in intact neurons is not possible using conventional voltage-clamp protocols, with the main barrier being the inability to accurately record Na_v_ channel activity in processes distant from the recording electrode. To circumvent this issue, we employed a protocol described by Milescu and colleagues [55] that uses a depolarizing pre-pulse step to inactivate Na_v_ channels distant from the recording electrode that is followed shortly afterward with a second step to record Na_v_ channels close to the recording pipette. Using this protocol, we reliably resolved well-clamped *I*_Na_ of intact pyramidal neurons in acute brain slice preparation. The same extracellular solution used for current-clamp recordings was also used to record *I*_Na_ with the addition of 120 µM CdCl_2_ (Sigma-Aldrich, St. Louis, MO, USA) to block Ca_2_^+^ currents. The intracellular solution for this recording contained the following (in mM): 100 Cs-gluconate (Hello Bio Inc., Princeton, NJ, USA); 10 tetraethylammonium chloride; 5 4-aminopyridine; 10 EGTA; 1 CaCl_2_; 10 HEPES; 4 Mg-ATP; 0.3 Na_3_-GTP; 4 Na_2_-phosphocreatine; and 4 NaCl (Sigma-Aldrich, St. Louis, MO, USA) (pH = 7.4 and osmolarity = 285 ± 5 mOsm/L; CsOH used to adjust pH and osmolarity). The same cocktail of synaptic blockers as the one used for the current-clamp recordings was used. Transient *I*_Na_ were evoked by using the voltage-clamp protocol shown in Figure 2 and as described elsewhere [55,56,81,82].

#### 4.4.3. Electrophysiological Data Analysis

In voltage-clamp experiments, the current densities were obtained by dividing the Na^+^ current (*I_Na_*) amplitude by membrane capacitance. Current–voltage relationships were generated by plotting current density as a function of the holding potential. Conductance (*G_Na_*) was calculated by the following Equation (1):(1)$$G_{Na}=\frac{I_{Na}}{\left(V_{m}-E_{rev}\right)}$$
where *I_Na_* is the current amplitude at voltage *V_m_*, and *E_rev_* is the Na^+^ reversal potential.

Activation curves were derived by plotting normalized *G_Na_* as a function of test potential and fitted using the Boltzmann equation. Equation (2):(2)$$\frac{G_{Na}}{G_{Na,Max}}=1+e^{\left\{\frac{\left(V_{a}-E_{m}\right)}{k}\right\}}$$
where *G_Na,Max_* is the maximum conductance, *V_a_* is the membrane potential of half-maximal activation, *E_m_* is the membrane voltage, and *k* is the slope factor. For steady-state inactivation, normalized current amplitude (*I_Na_*/*I_Na,Max_*) at the test potential was plotted as a function of pre-pulse potential (*V_m_*) and fitted using the Boltzmann equation. Equation (3):(3)$$\frac{I_{Na}}{I_{Na,Max}}=\frac{1}{\left\{1+e^{\frac{V_{h}-E_{m}}{k}}\right\}}$$
where *V_h_* is the potential of the half-maximal inactivation, and *k* is the slope factor.

In current-clamp recordings, the maximum number of APs was determined by quantifying the maximum number of APs a CA1 pyramidal cell fired at any current step during the evoked protocol (evoked APs were measured in response to a range of current injections from −20 to +210 pA, 800 ms). The current threshold (I_thr_) was defined as the current step at which at least one AP with overshoot was evoked (Figure 3C). The voltage threshold (V_thr_) was defined as the voltage at which the first-order derivative of the rising phase of the AP exceeded 10 mV/ms (from a plot of dV/dt versus V) (Figure 3D). Passive membrane properties, such as input resistance (R_in_) and membrane time constant (τ), were measured with current-clamp recordings with a membrane potential of −70 mV. To determine R_in_, the steady-state values of the voltage responses to a series of current steps from −120 to +20 pA with 20 pA increments/step and a duration of 500 ms were plotted as a voltage–current relationship. R_in_ was calculated as the slope of the data points fitted with linear regression. Membrane τ was calculated by fitting a single exponential function to the first 100–150 ms at a −40-pA hyperpolarizing, 500 ms current step from the same series. Cm was estimated as tau/R_in_ × gain of the amplifier (usually 1000) [52]. These data are also summarized in Table 3. Data analysis was performed using pClamp 9 (Clampfit 9) (Molecular Devices, Union City, CA, USA) and Origin 8.6 or Prism version 9.1.0, and the results were plotted with Origin 8.6 (OriginLab Corp., Northampton, MA, USA) or Prism version 9.1.0 (GraphPad Software, San Diego, CA, USA).

### 4.5. Immunohistochemistry

Following sectioning described above, rostro-caudally matched 300 µm slices of either hemisphere were treated with 20 µM triciribine (SelleckChem S1117) or 0.02% DMSO for 2 h in order to maximize the physiological changes that could be captured with immunohistochemistry and confocal imaging. Slices were then washed with 1× PBS for 5 min, fixed with 1% PFA with methanol (Sigma-Aldrich, St. Louis, MO, USA, #252549) for 30 min, and dehydrated in 20% sucrose overnight, as described previously [83]. The tissue was then embedded in OCT and sectioned at 30 µm and mounted onto SuperFrost Plus slides (Fisher Scientific 12-550-15). Tissue was washed in 1× PBS (3 times, 10 min each), blocked using a 10% NGS solution (Life Technologies, Carlsbad, CA, USA, #50062Z) for 60 min, and stained using the following primary antibodies overnight at 4 °C: anti-Na_v_1.6 msIgG1 (Antibodies Incorporated, Davis, CA, USA, #75-026), anti-AnkyrinG msIgG2a (NeuroMab 75–147), and anti-NeuN gpIgG (Synaptic System, Goettingen, Germany, #266-004). All of the primary antibodies were diluted at 1:300 in a solution of 3% BSA in PBS with 0.1% Tween-20. Following washes with 1× PBS (3 times, 10 min each), isotype specific Alexa Fluor antibodies (anti-msIgG1 488, Invitrogen #A21121, Carlsbad, CA, USA; anti-msIgG2a 568, Invitrogen A21,134; anti-gpIgG 647, Invitrogen #A21450) were diluted at 1:250 in a solution of 3% BSA in PBS with 0.1% Tween-20 and applied for 3 h at room temperature followed by 1× PBS washes (3 times, 10 min each), and they were then stained with 4′,6-diamidino-2-phenylindole (DAPI) for 5 min (Invitrogen D1306) and given a final 1× PBS wash (5 min). Tissues were then rinsed with ddH20 and dried for 10–15 min in a 30 °C oven before the coverslips were mounted using ProLong Gold (ThermoFisher #P36930, Waltham, MA, USA).

### 4.6. Confocal Imaging and Image Analysis

Confocal images were acquired with a Zeiss LSM-880 confocal microscope with a 63× oil immersion objective (1.4 NA). Multi-track acquisition was done with excitation lines at 405 for DAPI, 488 nm for Alexa 488, 561 nm for Alexa 568, and 633 nm for Alexa 647. Z-stacks were collected at z-steps of 0.43 µm for the first dataset, with a frame size of 1024 × 1024 and a pixel dwell time of 0.77 µs. All acquisition parameters, including photomultiplier gain and offset, were kept constant throughout each set of experiments. Acquired Z-stacks were sum-projected, and pixel intensity values were analyzed using Fiji ImageJ (fiji.sc). A total of four brain slices (*n* = 3 animals (DMSO) and *n* = 2 (triciribine)) were used for determining the length of Na_v_1.6 fluorescence intensity along the AIS in DMSO and triciribine treated conditions, and a single animal was used for the cross-section analysis of Na_v_1.6 staining at the AIS. For each condition, four z-stacks were acquired from each brain slice. With the experimenter blinded to the condition of each image, a region of interest (ROI) was highlighted on an overlay image of the somatic marker (NeuN) and the AIS marker (AnkyrinG) staining during the AnkyrinG positive processes. The “starting point” for the ROI was the point of reduced NeuN and increased AnkyrinG staining intensity, which corresponded to the AIS, as previously described [61]. The length of the AnkyrinG staining was considered the length of the AIS and was visually inspected to ensure that it corresponded to the length of Na_v_1.6 staining (*n* = 119 cells (DMSO) and 139 cells (triciribine)).

The cross-section analysis was performed as previously described [84]. Briefly, an ROI was made at the middle of the AIS to measure the fluorescent intensity perpendicular to the AIS, including 2.5 mm on either side of the peak AIS intensity using an ImageJ plugin LRoi (available at https://sites.imagej.net/CIPMM-MolPhys/, accessed on 3 January 2022) at the middle of the AIS. Fluorescent intensity was normalized to the background fluorescence (F_O_), which was calculated using the average fluorescent intensity of the first and last micron of the ROI (*n* = 16 cells (DMSO) and 20 cells (triciribine)). Data were tabulated in Excel and analyzed with GraphPad Prism 9. The linear filters available in Adobe Photoshop were applied to max projections of six optical slices from each condition for illustration purposes only.

### 4.7. Statistical Analysis

Results were expressed as mean ± standard error (SEM) or in box plots as Range (Perc 25, 75), Mean, Median, Min, and Max. The statistical significance of the observed differences among groups was determined by Student’s *t*-test. In case of the number of evoked action potentials, repeated-measure two-way ANOVA with uncorrected Fisher’s LSD was used.

For the immunohistochemistry, the statistical analysis was performed using a nested *t*-test to compare triciribine treatment to the DMSO control, with cells from individual slices that were considered to be technical replicates, with each bar in the figure representing a single brain slice (Figure 4M). For the cross-section analysis, a two-way ANOVA was used to compare DMSO to triciribine treatment (Figure 4N). The level of significance is listed in the figure legends for each experiment. Statistical analysis was performed using Origin 2021b software (OriginLab, Northampton, MA, USA) and GraphPad Prism version 9.1.0 (GraphPad Software, San Diego, CA, USA).

## Figures and Tables

**Figure 1 ijms-23-01700-f001:**
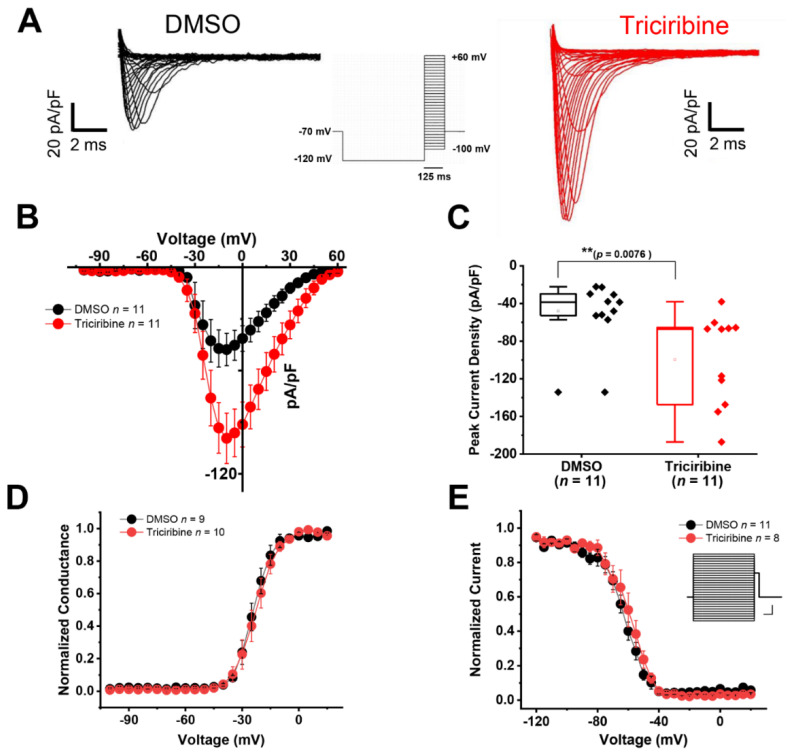
Pharmacological inhibition of Akt by triciribine leads to an increased *I*_Na_ current density in HEK-293 cells expressing Na_v_1.6. (**A**) Representative traces of *I*_Na_ currents recorded from HEK-Na_v_1.6 cells treated for 60 min either with DMSO (0.1%) (black) or 30 µM triciribine (red). (**B**) Current–voltage relationship showing increased peak *I*_Na_ current densities in triciribine-treated HEK-Na_v_1.6 cells (red, *n* = 11) compared to DMSO control (black, *n* = 11). (**C**) Peak current density at voltage step of −10 mV in DMSO and triciribine-treated HEK-Na_v_1.6 cells. A 60 min treatment with triciribine (30 µM) significantly increased the Na_v_1.6-mediated peak *I*_Na_ current density (*n* = 11 in both group; ** *p* = 0.0076 with Student’s unpaired *t*-test). (**D**,**E**). Voltage dependence of *I*_Na_ activation (**D**) and steady-state inactivation (**E**) in DMSO and triciribine-treated HEK-Na_v_1.6 cells. Treatment with triciribine did not affect voltage dependence of *I*_Na_ activation or steady-state inactivation. Inset shows the inactivation protocol (scale bars: 20 mv/100 ms). Box plot shows: Range (Perc 25, 75), Mean, Median, Min, and Max. Results are summarized in Table 1.

**Figure 2 ijms-23-01700-f002:**
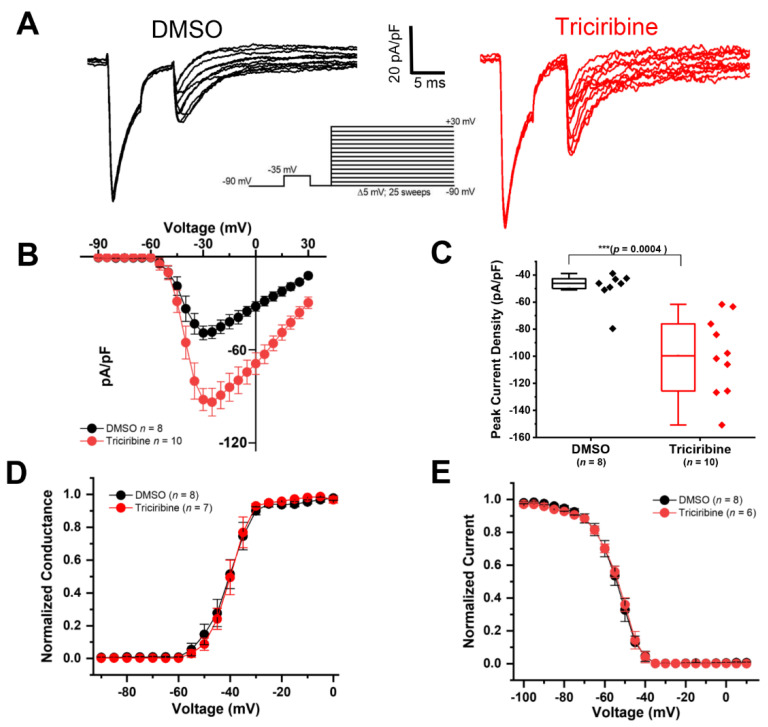
Pharmacological inhibition of Akt increases *I*_Na_ densities in CA1 hippocampal pyramidal neurons. (**A**) Representative traces of *I*_Na_ currents recorded from hippocampal neurons treated for 60 min either with DMSO (black) or triciribine 20 µM (red). (**B**) Current–voltage relationship showing increased peak *I*_Na_ current densities in triciribine-treated hippocampal pyramidal neurons. (**C**) Peak current density at voltage step of −10 mV in DMSO (*n* = 8) and triciribine-treated (*n* = 10) hippocampal neurons. A 60 min treatment with triciribine increased peak sodium current density (*** *p* = 0.0004 with Student’s *t*-test). (**D**,**E**) Voltage dependence of *I*_Na_ activation (**D**) and steady-state inactivation (**E**) of pyramidal cells from DMSO or triciribine-treated hippocampal slices. Treatment with triciribine did not affect neither the voltage dependence of *I*_Na_ activation nor steady-state inactivation. Box plot shows: Range (Perc 25, 75), Mean, Median, Min, and Max. Results are summarized in Table 2.

**Figure 3 ijms-23-01700-f003:**
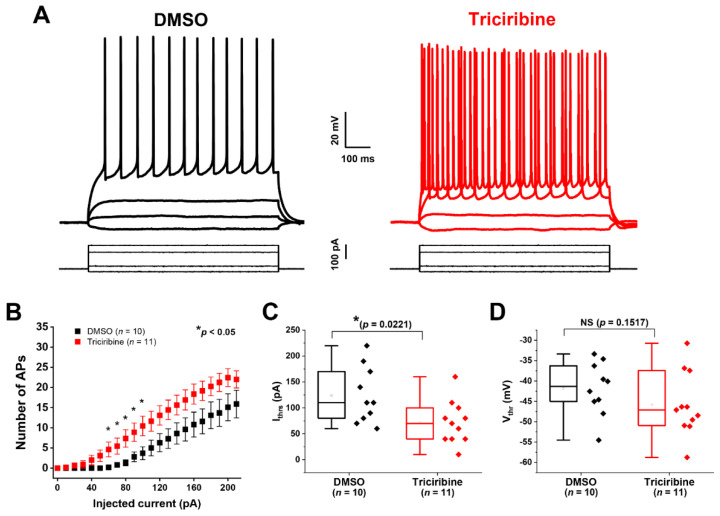
Akt inhibition increases excitability in CA1 hippocampal pyramidal neurons. (**A**) Representative traces showing trains of evoked action potentials recorded from CA1 pyramidal neurons treated for 60 min with either DMSO (black) or 20 µM triciribine (red) in response to increasing current injections from −20–210 pA (schematic of the current-clamp protocol is shown below representative traces). (**B**) Input–output curve showing increase in number of spikes in triciribine treated CA1 pyramidal neurons. Akt inhibition significantly increases the number of spikes across a broad range of injected currents. (*n* = 10 in DMSO and *n* = 11 in triciribine treated group; mean ± SEM, * *p* < 0.05 with Student’s *t*-test.) (**C**) Graph showing the effect of Akt inhibition on action potential current threshold. A 60 min treatment with triciribine significantly reduces the current threshold in CA1 pyramidal neurons. (*n* = 10 in DMSO and *n* = 11 in triciribine treated group; * *p* = 0.0221 with Student’s *t*-test). (**D**) Graph showing the effect of Akt inhibition on action potential voltage threshold. Treatment with triciribine did not affect voltage threshold in CA1 pyramidal neurons. *p* = 0.1517 with Student’s *t*-test. Box plot shows: Range (Perc 25, 75), Mean, Median, Min, and Max. Results with additional parameters are summarized in Table 3.

**Figure 4 ijms-23-01700-f004:**
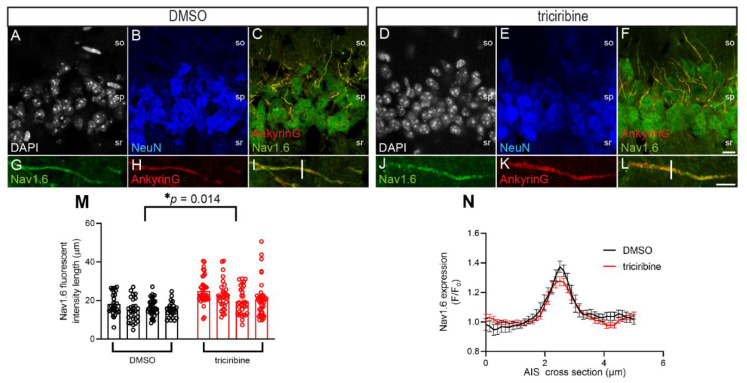
Treatment with triciribine alters length of the Na_v_1.6 immunofluorescence at the AIS but not fluorescent intensity of Na_v_1.6. Representative images of the CA1 region of (**A**–**C**) DMSO and (**D**–**F**) triciribine treated slices, with (**A**,**D**) DAPI (gray), (**B,E**) NeuN (blue), and (**C**,**F**) AnkyrinG (red) and Na_v_1.6 (green) merging. Scale bar in (**F**) is 10 µm. Zoom to AIS in (**G**–**I**) DMSO and (**J**–**L**) triciribine-treated slices with (**G**,**J**) Na_v_1.6 (green), (**H**,**K**) AnkyrinG (red), and (**I**,**L**) merging in both channels, and the white bar indicates an analyzed 5 µm cross-section. Scale bar in (**L**) indicates 5 µm. Stratum oriens (so), stratum pyramidale (sp), and stratum radiatum (sr) are indicated. (**M**) Treatment with 20 µM triciribine for 2 h significantly increased the length of the Na_v_1.6+ fluorescent intensity at the AIS, but (**N**) does not alter the fluorescent intensity of Na_v_1.6 as measured by the cross-section of the AIS. Data shown are mean ± SEM and separated scatter graph with bars to illustrate biological replicates (*n* = 4 brain slices per condition) in panel (**M**). * *p* = 0.0104 as analyzed by a two-tailed nested *t*-test. Data were log normalized before analysis. Zeiss 63× oil immersion objective (1.4 NA) was used in all cases.

**Table 1 ijms-23-01700-t001:** Effects of triciribine on Na_v_1.6 properties in HEK-Na_v_1.6 cells ^a^.

Treatment	Peak Density (pA/pF)	Activation V_1/2_ (mV)	k_act_ (mV)	Steady-State Inactivation V_1/2_ (mV)	k_inact_ (mV)
DMSO control (*n*)	−47.8 ± 9.4 (11)	−23.6 ± 1.2 (9)	3.5 ± 0.3 (9)	−62.9 ± 1.9 (11)	−6.6 ± 0.4 (11)
Triciribine 30 µM (*n*)	−99.4 ± 14.6 (11) **	−23.5 ± 2.1 (10)	3.6 ± 0.3 (10)	−59.5 ± 3.1 (8)	−5.8 ± 0.4 (8)

^a^ Summary of the electrophysiological evaluation of 20 µM triciribine in HEK-Na_v_1.6 cells. Results are expressed as mean ± SEM. The number of independent experiments is shown in parentheses. A Student’s *t*-test comparing cells treated with DMSO and 20 µM triciribine was used to determine statistical significance. ** *p* = 0.0076.

**Table 2 ijms-23-01700-t002:** Effects of triciribine on Na_v_ channel properties in CA1 pyramidal cells (slice preparation) ^a^.

Condition	Peak Density (pA/pF)	Activation V_1/2_ (mV)	k_act_ (mV)	Inactivation V_1/2_ (mV)	k_inact_ (mV)
DMSO control (*n*)	−49.5 ± 4.5 (8)	−41.4 ± 4.9 (8)	3.4 ± 0.7 (8)	−54.2 ± 3.5 (7)	−5.6 ± 0.7 (7)
Triciribine 20 µM (*n*)	−99. 4 ± 9.2 *** (10)	−40.3 ± 4.6 (10)	2.7 ± 0.3 (10)	−53.1 ± 2.3 (6)	−5.4 ± 0.3 (6)

^a^ Summary of the ex vivo voltage-clamp electrophysiology evaluation of triciribine in pyramidal cells of CA1. Results are expressed as mean ± SEM. The number of independent experiments is shown in parentheses. A Student’s unpaired *t*-test comparing pyramidal cells treated with 0.1% DMSO and 20 µM triciribine was used to determine statistical significance. ***, *p* = 0.0004.

**Table 3 ijms-23-01700-t003:** Effects of triciribine on active and passive properties of hippocampal CA1 pyramidal neurons ^a^.

Condition	RMP (mV)	V_trh_ (mV)	I_trh_ (pA)	R_in_ (MΩ)	Tau (ms)
DMSO (*n*)	−63.1 ± 1.6 (10)	−41.8 ± 2.1 (10)	124 ± 17.1 (10)	127.1 ± 10.7 (10)	14.7 ± 0.9 (10)
Triciribine 20 µM (*n*)	−64.7 ± 1.9 (11)	−45.8 ± 2.4 (11)	71.8 ± 12.5 * (11)	155.3 ± 9.7 (11)	16.9 ± 1.3 (11)

^a^ Summary of the ex vivo current-clamp electrophysiology evaluation of triciribine in pyramidal cells of CA1. Results are expressed as mean ± SEM. The number of independent experiments is shown in parentheses. A Student’s *t*-test comparing pyramidal cells treated with 0.1% DMSO and 20 µM triciribine was used to determine statistical significance. *, *p* = 0.0221 with Student’s *t*-test.

## Data Availability

Data included in this study are available upon request from the corresponding author.

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
