# Peer review of "Inhibition of the Akt/PKB Kinase Increases Nav1.6-Mediated Currents and Neuronal Excitability in CA1 Hippocampal Pyramidal Neurons"

_ijms, 2022, doi:10.3390/ijms23031700_

Round 1

Reviewer 1 Report

Marosi and colleagues present a paper entitled “Inhibition of the Akt/PKB kinase increases Nav1.6-mediated currents and neuronal excitability in CA1 hippocampal pyramidal neurons”.

The authors have tested the effect of triciribine, an inhibitor of Akt, on Nav currents in HEK-Nav1.6 cells and CA1 hippocampal pyramidal neurons, by using patch-clamp and immunohistochemistry techniques. They demonstrated that inhibition of Akt enhances peak INa density and the number of action potential firing by increasing the number of available Nav channels. In addition, triciribine induces an increase in the length of Nav1.6 at the axon initial segment.

The manuscript is well written, and a detailed electrophysiological analysis was conducted. In my opinion, this article responds to the scope of International Journal of Molecular Sciences and should be of interest and useful for its readership. Anyway, the manuscript requires minor revision, to be clearer and more exhaustive for the reader.

Major points

  1. In the first results section (2.1.) you test triciribine 30µM in HEK cells, then in the rest of the paper you test at 20µM concentration (also in tab1 you report 20µM). Why is this difference of concentration there?
  2. In the legend of figure 3A specify the current injection step of the traces and the description of this protocol (from -20 to 210pA?). in addition, traces at -20pA were never shown, so why are you start from -20pA?
  3. In figure 3B it is not clear to which points the asterisk refers.
  4. Why in immunohistochemistry the incubation is 2 hours instead of 1 hours?
  5. Specify the number of animals (N) used in brain slice electrophysiology.
  6. It is not clear if n=4 In all or for each condition and the number of animals (N) or slice (n).
  7. In the statistics section specify if the Student t-test is paired or unpaired and the post-test used in Anova analysis.

Minor points

  1. In figure 1B: fix -60 e -120.
  2. Add “n” of figure 1B, D and E (n of figure D and E are reported only in the text).
  3. Lines 137,138: replace in the text figure D,E with figure C,D.
  4. Lines 144-146: refer to fig 1E?
  5. Line 179: (legend tab. 2) “a Student t-test comparing” repeated two times.
  6. Line 192: specify if the INa is mesured at -10mV step.
  7. Line 207: write better the protocol (from -20-to +210 pA)
  8. Legend of figure 4: specify the objective used.
  9. Legend of figure 4: write M and N letters in bold
  10. Line 274: DAPI binds DNA of all cells, not only neurons, rephrase the sentence.
  11. Lines 386-389: specify how the HEK incubation is made (in the incubator, at room temperature,…).
  12. Line 403: paired-pulse term is not appropiate.
  13. Lines 403-405: insert this protocol inset in the figure.
  14. Lines 414-417: specify kind of the slice (coronal?).
  15. Lines 495-496: explain better the sentence.
  16. In the paper always write 1 hour or 60 minutes.
  17. In the tabs write (n) under the values.

Reviewer 2 Report

Overall, this is an interesting research article in which Marosi et al., investigate the effects of the Akt/PKB kinase inhibition on CA1 hippocampal neuronal excitability. By employing a combination of electrophysiology and immunohistochemistry they provide evidence that that Akt-mediated signal transduction affect neuronal excitability in Nav1.6-dependent manner.

However, I have several comments:

  1. Please check along the text the concentration of triciribine. Did you use 20 or 30 uM?
  2. Why did you use 1 hour treatment with triciribine for electrophysiological experiments and 2-hour treatment for Immunohistochemistry assays?
  3. What is the protocol in current clamp mode to measure current threshold? Did you measure it in long lasting current pulse? And how did you measure passive membranes properties? What are the protocols and pulses characteristic used for measure input resistance and time constant? In view of the relevance of these parameters in neuronal excitability these measures must be accurate, and I cannot appreciate it from the text. Overall, current clamp parameters are poorly described in method, you must improve this section in Material and Methods.
  4. What are you representing in Figure 3B? Is the I-F relationship for a representative cell with DMSO and triciribine? Have you represented the mean frequency for each current for the whole sample? I understand that if you performed statistical analysis, you are representing a mean, but you should indicate it in figure legends.
  5. Did you measure the firing frequency gain? It is a really good parameter to study neuronal excitability.
  6. You have state in the text that “Voltage responses and action potentials were evoked by hyper- and depolarizing current pulses (with a range of current injections from-20 pA to 210 pA with 800 ms pulses and a change in injected current of 10 pA between sweeps) from a starting holding membrane potential of −70 mV”. In current clamp mode it doesn’t make sense to fix the membrane potential and you must start the experiment from the real membrane potential of the cell. Is this a mistake or did you injected current to drive the membrane potential to -70 mV before starting the experiment?
  7. Other minor corrections:

Line 57. Although CNS is a well know abbreviature, please use the word in full when you refer for the first time.

The word “triciribine” appears in the text, figures, and figure legends sometimes with capital letters and others in lowercase, put them at least in the text all in lowercase.

Line 212, change neuros to neuron

Line 274 change Figure 4 A to Figure 4A

Reviewer 3 Report

In this manuscript, the authors describe the change of excitability by Akt/PKB kinase inhibition in CA1 hippocampal pyramidal neurons. They showed that triciribine treatment for a couple of hours potentiated neuronal voltage-dependent Na+ channel activity in Nav1.6 expressing HEK cells and in CA1 hippocampal pyramidal neurons. In addition, they found that the triciribine treatment altered the localization of Nav1.6 assayed with immunofluorescence labelling at AIS. Overall, the manuscript was well written and experimental results were solid except some points. Following concerns should be addressed for improvement.

  1. The concentration of triciribine was inconsistent. In Table1, 20 uM triciribine was used. However, in text (lines 116, 133), 30 uM was shown.

  1. The activation property of voltage-dependent Na+ channel currents were different between Nav1.6 expressing HEK cells and CA1 hippocampal pyramidal neurons, in particular the difference of V1/2 was around 20mV. This difference is too big to conclude that the same type of channels is involved, hence implying that other inward currents except Nav1.6 was activated by depolarization in your experimental protocol.

  1. The explanation of statistical analysis is confusing. The use of ANOVA is not clear (lines557-558). Two way (treatment and injected current) ANOVA but not t-test should be applied to Fig.3B. The analysis of Fig.4M is not clear. Please explain the four sets of columns in DMSO and triciribine.

  1. Judging from the pipette solution for current-clamp experiments (lines 463-466), K+ currents were blocked by Cs+, TEA, and 4-AP. This experimental protocol is unusual to record neuronal membrane excitability such as action potentials, resting membrane potential, input resistance, and local response show in Table 3.
